# Accurate autocorrelation modeling substantially improves fMRI reliability

Wiktor Olszowy [1,2], John Aston[3], Catarina Rua[1] & Guy B. Williams[1]

Given the recent controversies in some neuroimaging statistical methods, we compare the most frequently used functional Magnetic Resonance Imaging (fMRI) analysis packages: AFNI, FSL and SPM, with regard to temporal autocorrelation modeling. This process, sometimes known as pre-whitening, is conducted in virtually all task fMRI studies. Here, we employ eleven datasets containing 980 scans corresponding to different fMRI protocols and subject populations. We found that autocorrelation modeling in AFNI, although imperfect, performed much better than the autocorrelation modeling of FSL and SPM. The presence of residual autocorrelated noise in FSL and SPM leads to heavily confounded first level results, particularly for low-frequency experimental designs. SPM's alternative pre-whitening method, FAST, performed better than SPM's default. The reliability of task fMRI studies could be improved with more accurate autocorrelation modeling. We recommend that fMRI analysis packages provide diagnostic plots to make users aware of any pre-whitening problems.

[1] Wolfson Brain Imaging Centre, Department of Clinical Neurosciences, University of Cambridge, Cambridge CB2 0QQ, UK. [2] Laboratory of Research in Neuroimaging (LREN), Department of Clinical Neurosciences, CHUV, University of Lausanne, 1011 Lausanne, Switzerland. [3] Statistical Laboratory, Department of Pure Mathematics and Mathematical Statistics, University of Cambridge, Cambridge CB3 0WB, UK. Correspondence and requests for materials should be addressed to W.O. (email: olszowyw@gmail.com)

Functional magnetic resonance imaging (fMRI) data are known to be positively autocorrelated in time[1]. It results from neural and hemodynamic sources, but also from scanner-induced low-frequency drifts, respiration, and cardiac pulsation, as well as from movement artifacts not accounted for by motion correction[2]. If this autocorrelation is not accounted for, spuriously high fMRI signal at one time point can be prolonged to the subsequent time points, which increases the likelihood of obtaining false positives in task studies[3]. As a result, parts of the brain might erroneously appear active during an experiment. The degree of temporal autocorrelation is different across the brain[4]. In particular, autocorrelation in gray matter is stronger than in white matter and cerebrospinal fluid, but it also varies within gray matter.

AFNI[5], FSL[6], and SPM[7], the most popular packages used in fMRI research, first remove the signal at very low frequencies (for example using a high-pass filter), after which the residual temporal autocorrelation is estimated from the residuals of an initial Ordinary Least Squares (OLS) model and is removed in a process called pre-whitening. In AFNI temporal autocorrelation is modeled voxel-wise. For each voxel, an autoregressive-moving-average ARMA(1,1) model is estimated[8]. The two ARMA(1,1) parameters are estimated only on a discrete grid and are not spatially smoothed. For FSL, a Tukey taper is used to smooth the spectral density estimates voxel-wise. These smoothed estimates are then additionally smoothed within tissue type. Woolrich et al.[9] has shown the applicability of the FSL's method in two fMRI protocols: with repetition time (TR) of 1.5 and of 3 s, and with voxel size $4 \times 4 \times 7$ mm$^3$. By default, SPM estimates temporal autocorrelation globally as an autoregressive AR(1) plus white noise process[10]. SPM has an alternative approach: FAST, but we know of only three studies, which have used it[11–13]. FAST uses a dictionary of covariance components based on exponential covariance functions[13]. More specifically, the dictionary is of length $3p$ and is composed of $p$ different exponential time constants along their first and second derivatives. By default, FAST employs 18 components. Like SPM's default pre-whitening method, FAST is based on a global noise model.

Lenoski et al.[14] compared several fMRI autocorrelation modeling approaches for one fMRI protocol (TR = 3 s, voxel size $3.75 \times 3.75 \times 4$ mm$^3$). The authors found that the use of the global AR(1), of the spatially smoothed AR(1) and of the spatially smoothed FSL-like noise models resulted in worse whitening performance than the use of the non-spatially smoothed noise models. Eklund et al.[15] showed that in SPM the shorter the TR, the more likely it is to get false positive results in first-level (also known as single subject) analyses. It was argued that SPM often does not remove a substantial part of the autocorrelated noise. The relationship between shorter TR and increased false positive rates was also shown for the case when autocorrelation is not accounted[3].

In this study we investigate the whitening performance of AFNI, FSL, and SPM for a wide variety of fMRI blood-oxygen-level-dependent protocols. We analyze both the default SPM's method and the alternative one: FAST. Furthermore, we analyze the resulting specificity-sensitivity trade-offs in first-level fMRI results, and investigate the impact of pre-whitening on second-level analyses. We observe better whitening performance for AFNI and SPM tested with option FAST than for FSL and SPM. Imperfect pre-whitening heavily confounds first-level analyses.

## Results

### Whitening performance of AFNI, FSL, and SPM.
To investigate the whitening performance resulting from the use of noise models in AFNI, FSL, and SPM, we plotted the power spectra of the general linear model (GLM) residuals. Figure 1 shows the power spectra averaged across all brain voxels and subjects for smoothing of 8 mm and assumed boxcar design of 10 s of rest followed by 10 s of stimulus presentation. The statistical inference in AFNI, FSL, and SPM relies on the assumption that the residuals after pre-whitening are white. For white residuals, the power spectra should be flat. However, for all the datasets and all the packages, there was some visible structure. The strongest artifacts were visible for FSL and SPM at low frequencies. At high frequencies, power spectra from FAST were closer to 1 than power spectra from the other methods. Figure 1 does not show respiratory spikes, which one could expect to see. This is because the figure refers to averages across subjects. We observed respiratory spikes when analyzing power spectra for single subjects (not shown).

### Resulting specificity-sensitivity trade-offs.
In order to investigate the impact of the whitening performance on first-level results, we analyzed the spatial distribution of significant clusters in AFNI, FSL, and SPM. Figure 2 shows an exemplary axial slice in the Montreal Neurological Institute (MNI) space for 8 mm smoothing. It was made through the imposition of subjects' binarized significance masks on each other. Scale refers to the percentage of subjects within a dataset where significant activation was detected at the given voxel. The x-axis corresponds to four assumed designs. Resting state data were used as null data. Thus, low numbers of significant voxels were a desirable outcome, as this was suggesting high specificity. Task data with assumed wrong designs were used as null data too. Thus, clear differences between the true design (indicated with red boxes) and the wrong designs were a desirable outcome. For FSL and SPM, often the relationship between lower assumed design frequency (boxcar40 vs. boxcar12) and an increased number of significant voxels was visible, in particular for the resting state datasets: "FCP Beijing", "FCP Cambridge", and "Cambridge Research into Impaired Consciousness (CRIC)". For null data, significant clusters in AFNI were scattered primarily within gray matter. For FSL and SPM, many significant clusters were found in the posterior cingulate cortex, while most of the remaining significant clusters were scattered within gray matter across the brain. False positives in gray matter occur due to the stronger positive autocorrelation in this tissue type compared to white matter[4]. For the task datasets: "NKI checkerboard TR = 1.4 s", "NKI checkerboard TR = 0.645 s", "BMMR checkerboard", and "CRIC checkerboard" tested with the true designs, the majority of significant clusters were located in the visual cortex. This resulted from the use of visual experimental designs for the fMRI task. For the impaired consciousness patients (CRIC), the registrations to MNI space were imperfect, as the brains were often deformed.

### Additional comparison approaches.
The above analysis referred to the spatial distribution of significant clusters on an exemplary axial slice. As the results can be confounded by the comparison approach, we additionally investigated two other comparison approaches: the percentage of significant voxels and the positive rate. Supplementary Fig. 1 shows the average percentage of significant voxels across subjects in 10 datasets for smoothing of 8 mm and for 16 assumed boxcar experimental designs. As more designs were considered, the relationship between lower assumed design frequency and an increased percentage of significant voxels in FSL and SPM (discussed before for Fig. 2) was even more apparent. This relationship was particularly interesting for the "CRIC checkerboard" dataset. When tested with the true design, the percentage of significant voxels for AFNI, FSL, SPM, and FAST was similar: 1.2, 1.2, 1.5, and 1.3%, respectively.

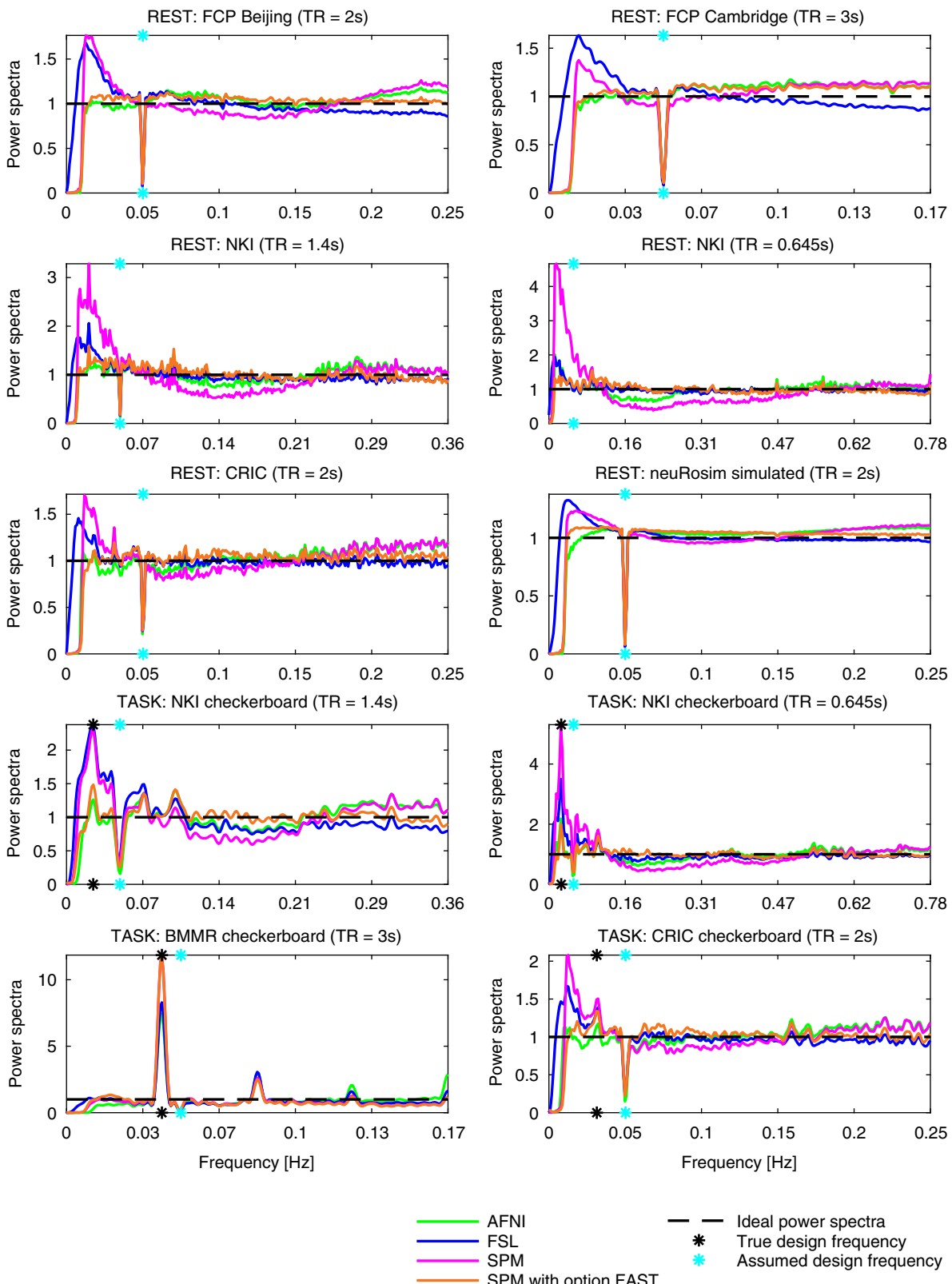

**Fig. 1** Power spectra of the GLM residuals in native space averaged across brain voxels and across subjects for the assumed boxcar design of 10 s of rest followed by 10 s of stimulus presentation (boxcar10). The dips at 0.05 Hz are due to the assumed design period being 20 s (10 s + 10 s). For some datasets, the dip is not seen as the assumed design frequency was not covered by any of the sampled frequencies. The frequencies on the x-axis go up to the Nyquist frequency, which is 0.5/repetition time. If after pre-whitening the residuals were white (as it is assumed), the power spectra would be flat. AFNI and SPM's alternative method: FAST, led to best whitening performance (most flat spectra). For FSL and SPM, there was substantial autocorrelated noise left after pre-whitening, particularly at low frequencies

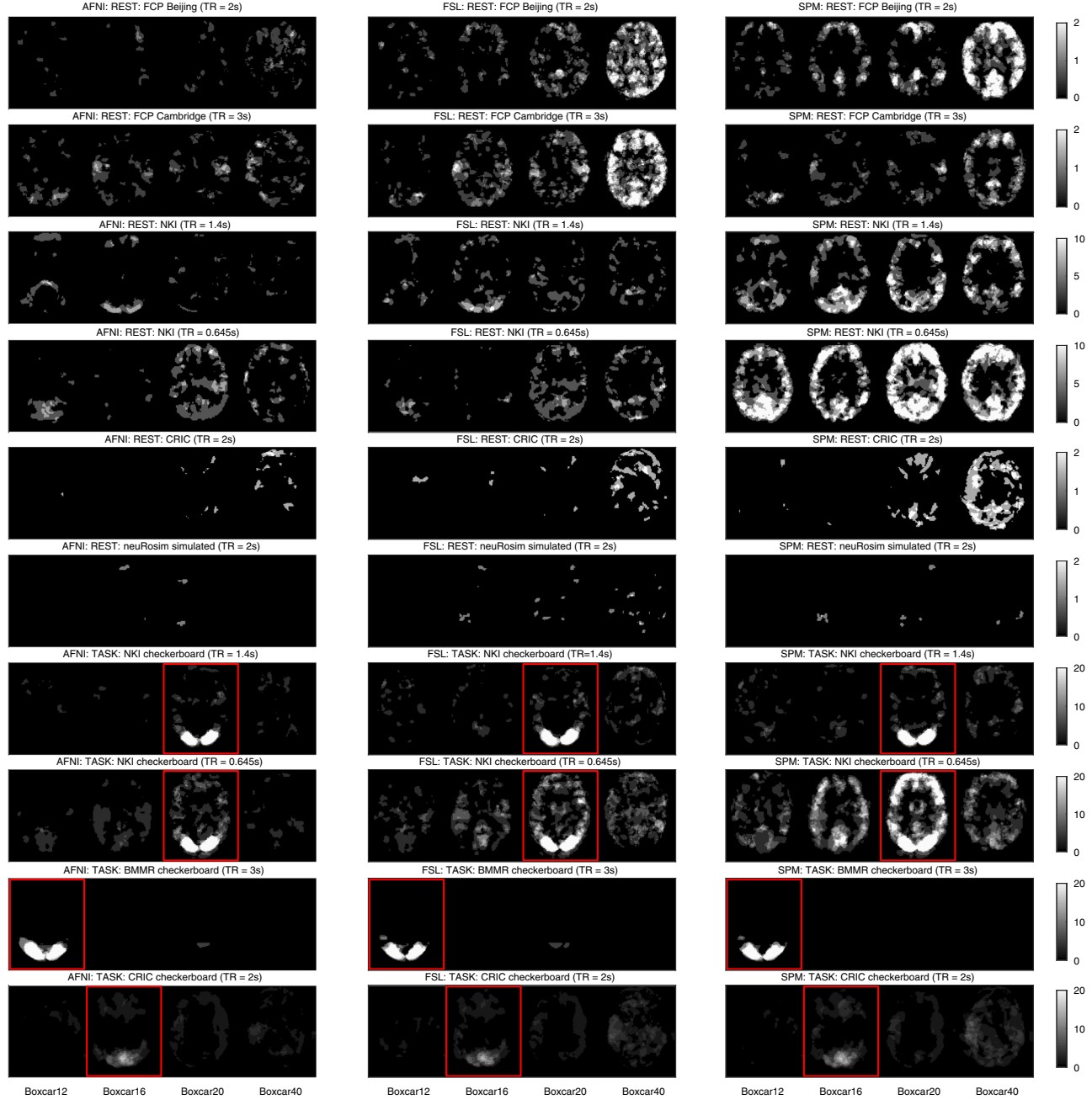

**Fig. 2** Spatial distribution of significant clusters in AFNI (left), FSL (middle), and SPM (right) for different assumed experimental designs. Scale refers to the percentage of subjects where significant activation was detected at the given voxel. The red boxes indicate the true designs (for task data). Resting state data were used as null data. Thus, low numbers of significant voxels were a desirable outcome, as it was suggesting high specificity. Task data with assumed wrong designs were used as null data too. Thus, large positive differences between the true design and the wrong designs were a desirable outcome. The clearest cut between the true and the wrong/dummy designs was obtained with AFNI's noise model. FAST performed similarly to AFNI's noise model (not shown)

However, AFNI and FAST returned much lower percentages of significant voxels for the assumed wrong designs. For the assumed wrong design "40", FSL and SPM returned on average a higher percentage of significant voxels than for the true design: 1.4 and 2.2%, respectively. Results for AFNI and FAST for the same design showed only 0.3 and 0.4% of significantly active voxels.

Overall, at an 8 mm smoothing level, AFNI and FAST outperformed FSL and SPM showing a lower average percentage of significant voxels in tests with the wrong designs: on average across 10 datasets and across the wrong designs, the average percentage of significant voxels was 0.4% for AFNI, 0.9% for FSL, 1.9% for SPM, and 0.4% for FAST.

As multiple comparison correction depends on the smoothness level of the residual maps, we also checked the corresponding differences between AFNI, FSL, and SPM. The residual maps seemed to be similarly smooth. At an 8 mm smoothing level, the average geometric mean of the estimated full width at half maxima of the Gaussian distribution in $x$-, $y$-, and $z$-dimensions across the 10 datasets and across the 16 assumed designs was

10.9 mm for AFNI, 10.3 mm for FSL, 12.0 mm for SPM, and 11.8 mm for FAST. Moreover, we investigated the percentage of voxels with $z$-statistic above 3.09. This value is the 99.9% quantile of the standard normal distribution and is often used as the cluster defining threshold. For null data, this percentage should be 0.1%. The average percentage across the 10 datasets and across the wrong designs was 0.6% for AFNI, 1.2% for FSL, 2.1% for SPM, and 0.4% for FAST.

Supplementary Figs. 2, 3 show the positive rate for smoothing of 4 and 8 mm. The general patterns resemble those already discussed for the percentage of significant voxels, with AFNI and FAST consistently returning lowest positive rates (familywise error rates, FWER) for resting state scans and task scans tested with wrong designs. For task scans tested with the true designs, the positive rates for the different pre-whitening methods were similar. The black horizontal lines show the 5% false positive rate, which is the expected proportion of scans with at least one significant cluster if in reality there was no experimentally induced signal in any of the subjects' brains. The dashed horizontal lines are the confidence intervals for the proportion of false positives. These were calculated knowing that variance of a Bernoulli($p$) distributed random variable is $p(1 − p)$. Thus, the confidence intervals were $0.05 \pm \sqrt{0.05 \cdot 0.95/n}$, with $n$ denoting the number of subjects in the dataset.

Since smoothing implicitly affects the voxel size, we considered different smoothing kernel sizes. We chose 4, 5, and 8 mm, as these are the defaults in AFNI, FSL, and SPM. No smoothing was also considered, as for 7T data this preprocessing step is sometimes avoided[16,17]. With a wider smoothing kernel, the percentage of significant voxels increased (not shown), while the positive rate decreased (Supplementary Figs. 2, 3). Differences between AFNI, FSL, SPM, and FAST discussed above for the four comparison approaches and smoothing of 8 mm were consistent across the four smoothing levels.

Further results are available from https://github.com/wiktorolszowy/fMRI_temporal_autocorrelation/tree/master/figures

**Event-related design studies**. In order to check if differences in autocorrelation modeling in AFNI, FSL, and SPM lead to different first-level results for event-related design studies, we analyzed the CamCAN dataset. The task was a sensorimotor one with visual and audio stimuli, to which the participants responded by pressing a button. The design was based on an m-sequence[18]. Supplementary Fig. 4 shows (1) power spectra of the GLM residuals in native space averaged across brain voxels and across subjects for the assumed true design (E1), (2) average percentage of significant voxels for three wrong designs and the true design, (3) positive rate for the same four designs, and (4) spatial distribution of significant clusters for the assumed true design (E1). Only smoothing of 8 mm was considered. The dummy event-related design (E2) consisted of relative stimulus onset times generated from a uniform distribution with limits 3 and 6 s. The stimulus duration times were 0.1 s.

For the assumed low-frequency design (B2), AFNI's autocorrelation modeling led to the lowest familywise error rate as residuals from FSL and SPM again showed a lot of signal at low frequencies. However, residuals from SPM tested with option FAST were similar at low frequencies to AFNI's residuals. As a result, the familywise error rate was similar to AFNI. For high frequencies, power spectra from SPM tested with option FAST were more closely around 1 than power spectra corresponding to the standard three approaches (AFNI/FSL/SPM). For an event-related design with very short stimulus duration times (around zero), residual positive autocorrelation at high frequencies makes it difficult to distinguish the activation blocks from the rest

blocks, as part of the experimentally induced signal is in the assumed rest blocks. This is what happened with AFNI and SPM. As their power spectra at high frequencies were above 1, we observed for the true design a lower percentage of significant voxels compared to SPM tested with option FAST. On the other hand, FSL's power spectra at high frequencies were below 1. As a result, FSL decorrelated activation blocks from rest blocks possibly introducing negative autocorrelations at high frequencies, leading to a higher percentage of significant voxels than SPM tested with option FAST. Though we do not know the ground truth, we might expect that AFNI and SPM led for this event-related design dataset to more false negatives than SPM with option FAST, while FSL led to more false positives. Alternatively, FSL might have increased the statistic values above their nominal values for the truly but little active voxels.

**Slice timing correction**. As slice timing correction is an established preprocessing step, which often increases sensitivity[19], we analyzed its impact on pre-whitening for two datasets for which we knew the acquisition order: "CRIC checkerboard" and "CamCAN sensorimotor". "CRIC checkerboard" scans were acquired with an interleave acquisition starting with the second axial slice from the bottom (followed with fourth slice, etc.), while "CamCAN sensorimotor" scans were acquired with a descending acquisition with the most upper axial slice being scanned first. We considered only the true designs. For the two datasets and for the four pre-whitening methods, slice timing correction changed the power spectra of the GLM residuals in a very limited way (Supplementary Fig. 5). Regardless of whether slice timing correction was performed or not, pre-whitening approaches from FSL and SPM left substantial positive autocorrelated noise at low frequencies, while FAST led to even more flat power spectra than AFNI. We also investigated the average percentage of significant voxels (Supplementary Table 1). Slice timing correction changed the amount of significant activation only negligibly, with the exception of AFNI's pre-whitening in the "CamCAN sensorimotor" scans. In the latter case, the apparent sensitivity increase (from 7.64 to 13.45% of the brain covered by significant clusters) was accompanied by power spectra of the GLM residuals falling below 1 for the highest frequencies. This suggests negative autocorrelations were introduced at these frequencies, which could have led to statistic values being on average above their nominal values.

**Group studies**. To investigate the impact of pre-whitening on the group level, we performed via SPM summary statistic analyses, and via AFNI's 3dMEMA[8] we performed mixed effects analyses. To be consistent with a previous study on group analyses[20], we considered one-sample $t$-test with sample size 20. For each dataset, we considered the first 20 subjects. We exported coefficient maps and $t$-statistic maps (from which standard errors can be derived) following 8 mm spatial smoothing and pre-whitening from AFNI, FSL, SPM, and FAST. Both for the summary statistic analyses and for the mixed effects analyses, we employed cluster inference with cluster defining threshold of 0.001 and significance level of 5%. Altogether, we performed 1312 group analyses: 2 (for summary statistic/mixed) × 4 (for pre-whitening) × (10 × 16 + 4) (for the first 10 datasets tested with 16 boxcar designs each and for the eleventh dataset tested with four designs). We found significant activation for 236 analyses, which we listed in Supplementary Data.

For each combination of group analysis model and pre-whitening (2 × 4), we ran 164 analyses. As five datasets were task datasets, 159 analyses ran on null data. Supplementary Table 2 shows FWER for the summary statistic and mixed effects null

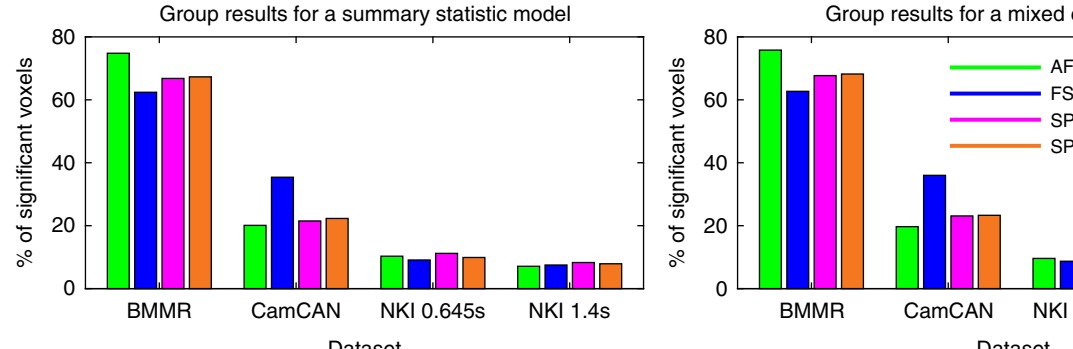

**Fig. 3** Group results for four task datasets with assumed true designs. Summary statistic analyses and mixed effects analyses led to only negligibly different percentages of significant voxels

data analyses, and for the four pre-whitening approaches. On average, FWER for the mixed effects analyses was almost twice higher than FWER for the summary statistic analyses. The use of AFNI's pre-whitening led to highest FWER, while FAST led to lower FWER than the SPM's default approach.

Figure 3 shows the percentage of significant voxels for four task datasets with assumed true designs. Results for the "CRIC checkerboard" dataset are not shown, as no significant clusters were found at the group level. This occurred due to several of the subjects having deformed brains, which led to the group brain mask not covering the primary visual cortex. For the "BMMR checkerboard" dataset, the brain mask was limited mainly to the occipital lobe and the percentage relates to the field of view that was used. Both for the summary statistic analyses and for the mixed effects analyses, we observed little effect of pre-whitening. For task data tested with true designs, we found only negligible differences between the summary statistic analyses and the mixed effects analyses.

Noteworthily, for the event-related task dataset "CamCAN sensorimotor" tested with the true design, the use of FAST led to slightly higher amount of significant activation compared to the default SPM's method, while FSL led to much higher amount of significant activation. This means that for this event-related design dataset, the sensitivity differences from the first-level analyses propagated to the second level. This happened both for the summary statistic model and for the mixed effects model.

### Discussion

In the case of FSL and SPM for the datasets "FCP Beijing", "FCP Cambridge", "CRIC RS", and "CRIC checkerboard", there was a clear relationship between lower assumed design frequency and an increased percentage of significant voxels. This relationship exists when positive autocorrelation is not removed from the data[3]. Autocorrelated processes show increasing variances at lower frequencies. Thus, when the frequency of the assumed design decreases, the mismatch between the true autocorrelated residual variance and the incorrectly estimated white noise variance grows. In this mismatch, the variance is underestimated, which results in a larger number of false positives.

An interesting case was the checkerboard experiment conducted with impaired consciousness patients, where FSL and SPM found a higher percentage of significant voxels for the design with the assumed lowest design frequency than for the true design. As this subject population was unusual, one might suspect weaker or inconsistent response to the stimulus. However, positive rates for this experiment for the true design were all around 50%, substantially above other assumed designs.

Compared to FSL and SPM, the use of AFNI's and FAST noise models for task datasets resulted in larger differences between the

true design and the wrong designs in the first-level results. This occurred because of more accurate autocorrelation modeling in AFNI and in FAST. In our analyses, FSL and SPM left a substantial part of the autocorrelated noise in the data and the statistics were biased. For none of the pre-whitening approaches were the positive rates around 5%, which was the significance level used in the cluster inference. This is likely due to imperfect cluster inference in FSL. High familywise error rates in first-level FSL analyses were already reported[21]. In our study the familywise error rate following the use of AFNI's and FAST noise models was consistently lower than the familywise error rate following the use of FSL's and SPM's noise models. Opposed to the average percentage of significant voxels, high familywise error rate directly points to problems in the modeling of many subjects.

The highly significant responses for the Nathan Kline Institute (NKI) datasets are in line with previous findings[15], where it was shown that for fMRI scans with short TR it is more likely to detect significant activation. The NKI scans that we considered had TR of 0.645 and 1.4 s, in both cases much shorter than the usual TRs. Such short TRs are now possible due to multiband sequences[22]. The shorter the TR the higher the correlations between adjacent time points[3]. If positive autocorrelation in the data is higher than the estimated level, then false positive rates will increase. The former study[15] only referred to SPM. In addition to the previous study, we observed that the familywise error rate for short TRs was substantially lower in FSL than in SPM, though still much higher than for resting state scans at TR = 2 s ("FCP Beijing" and "CRIC RS"). FSL models autocorrelation more flexibly than SPM, which seems to be confirmed by our study. For short TRs, AFNI's performance deteriorated too, as autocorrelation spans much more than one TR and an ARMA (1,1) noise model can only partially capture it.

Apart from the different TRs, we analyzed the impact of spatial smoothing. If more smoothing is applied, the signal from gray matter will be often mixed with the signal from white matter. As autocorrelation in white matter is lower than in gray matter[4], autocorrelation in a primarily gray matter voxel will likely decrease following stronger smoothing. The observed relationships of the percentage of significant voxels and of the positive rate from the smoothing level can be surprising, as random field theory is believed to account for different levels of data smoothness. The relationship for the positive rate (familywise error rate) was already known[15,21]. The impact of smoothing and spatial resolution was investigated in a number of previous studies[23–25]. We considered smoothing only as a confounder. Importantly, for all four levels of smoothing, AFNI and FAST outperformed FSL and SPM.

Our results confirm Lenoski et al.[14], insofar as our study also showed problems with SPM's default pre-whitening. Interestingly,

Eklund et al.[21] already compared AFNI, FSL, and SPM in the context of first-level fMRI analyses. AFNI resulted in substantially lower false positive rates than FSL and slightly lower false positive rates than SPM. We also observed lowest false positive rates for AFNI. Opposed to that study[21], which compared the packages in their entirety, we compared the packages only with regard to pre-whitening. It is possible that pre-whitening is the most crucial single difference between AFNI, FSL, and SPM, and that the relationships described by Eklund et al.[21] would look completely different if AFNI, FSL, and SPM employed the same pre-whitening. For one dataset, Eklund et al.[21] also observed that SPM led to worst whitening performance.

The differences in first-level results between AFNI, FSL, and SPM, which we found could have been smaller if physiological recordings had been modeled. The modeling of physiological noise is known to improve whitening performance, particularly for short TRs[2,12,13]. Unfortunately, cardiac and respiratory signals are not always acquired in fMRI studies. Even less often are the physiological recordings incorporated to the analysis pipeline. Interestingly, a recent report suggested that the FSL's tool ICA FIX applied to task data can successfully remove most of the physiological noise[26]. This was shown to lower the familywise error rate at the group level compared to previous findings[20]. Such an approach corresponds to more accurate pre-whitening. However, in our analyses the different pre-whitening methods affected the group level analyses only in a very negligible way. While Eklund et al.[20] found that group-level analyses are strongly confounded by spatial autocorrelation modeling, we found that single subject analyses were strongly confounded by the pre-whitening accuracy.

In our main analysis pipeline we did not perform slice timing correction. For two datasets, we additionally considered slice timing correction and observed very similar first-level results compared to the case without slice timing correction. The observed little effect of slice timing correction is likely a result of the temporal derivative being modeled within the GLM framework. This way a large part of the slice timing variation might have been captured without specifying the exact slice timing. For the only case where slice timing correction led to noticeably higher amount of significant activation, we observed negative autocorrelations at high frequencies in the GLM residuals. If one did not see the power spectra of the GLM residuals, slice timing correction in this case could be thought to directly increase sensitivity, while in fact pre-whitening confounded the comparison.

FSL is the only package with a benchmarking paper of its pre-whitening approach[9]. The study employed data corresponding to two fMRI protocols. For one protocol TR was 1.5 s, while for the other protocol TR was 3 s. For both protocols, the voxel size was $4 \times 4 \times 7$ mm$^3$. These were large voxels. We suspect that the FSL's pre-whitening approach could have been overfitted to this data. Regarding SPM, pre-whitening with simple global noise models was found to result in profound bias in at least two previous studies[14,27]. SPM's default is a simple global noise model. However, SPM's problems could be partially related to the estimation procedure. First, the estimation is approximative as it uses a Taylor expansion[10]. Second, the estimation is based on a subset of the voxels. Only voxels with $p < 0.001$ following inference with no pre-whitening are selected. This means that the estimation strongly depends both on the TR and on the experimental design[3].

If the second-level analysis is performed with a summary statistic model, the standard error maps are not used. Thus, standard models like the summary statistic approach in SPM should not be affected by imperfect pre-whitening[28]. On the other hand, residual positive autocorrelated noise decreases the signal differences

between the activation blocks and the rest blocks. This is relevant for event-related designs. Bias from confounded coefficient maps can be expected to propagate to the group level. We showed that pre-whitening indeed confounds group analyses performed with a summary statistic model. However, more relevant is the case of mixed effects analyses, for example when using 3dMEMA in AFNI[8] or FLAME in FSL[29]. These approaches additionally employ standard error maps, which are directly confounded by imperfect pre-whitening. Bias in mixed effects fMRI analyses resulting from non-white noise at the first level was already reported[30]. Surprisingly, we did not observe pre-whitening-induced specificity problems for analyses using 3dMEMA, including for very short TRs. Importantly, this means that imperfect pre-whitening does not meaningfully affect group results when using 3dMEMA. It indicates that the between-subject variability in the considered datasets was negligible compared to the within-subject variability, and that 3dMEMA operates on the ratio of the within-subject variability to the between-subject variability rather than on the absolute variability values.

For task datasets tested with true designs, the results from summary statistic analyses differed very little compared to 3dMEMA results. FLAME was also shown to have similar sensitivity compared to summary statistic analyses[31]. However, mixed effects models should be more optimal than summary statistic models as they employ more information. Although group analysis modeling in task fMRI studies needs to be investigated further, it is beyond the scope of this paper. As mixed effects models employ standard error maps, bias in them should be avoided.

Problematically, for resting state data treated as task data, it is possible to observe activation both in the posterior cingulate cortex and in the frontal cortex, since these regions belong to the default mode network[32]. In fact, in Supplementary Fig. 18 in Eklund et al.[20] the spatial distribution plots of significant clusters indicate that the significant clusters appeared mainly in the posterior cingulate cortex, even though the assumed design for that analysis was a randomized event-related design. The rest activity in these regions can occur at different frequencies and can underlie different patterns[33]. Thus, resting state data are not perfect null data for task fMRI analyses, especially if one uses an approach where a subject with one small cluster in the posterior cingulate cortex enters an analysis with the same weight as a subject with a number of large clusters spread throughout the entire brain. Task fMRI data are not perfect null data either, as an assumed wrong design might be confounded by the underlying true design. For simulated data, a consensus is needed how to model autocorrelation, spatial dependencies, physiological noise, scanner-dependent low-frequency drifts, and head motion. Some of the current simulation toolboxes[34] enable the modeling of all these aspects of fMRI data, but as the later analyses might heavily depend on the specific choice of parameters, more work is needed to understand how the different sources of noise influence each other. In our study, results for simulated resting state data were substantially different compared to acquired real resting state scans. In particular, the percentage of significant voxels for the simulated data was much lower, indicating that the simulated data did not appropriately correspond to the underlying brain physiology. Considering resting state data where the posterior cingulate cortex and the frontal cortex are masked out could be an alternative null. Because there is no perfect fMRI null data, we used both resting state data with assumed dummy designs and task data with assumed wrong designs. Results for both approaches coincided.

Unfortunately, although the vast majority of task fMRI analyses is conducted with linear regression, the popular analysis packages do not provide diagnostic plots. For old versions of SPM, the

**Table 1 Overview of the employed datasets**

| Study | Experiment | Place | Design | No. subjects | Field [T] | TR [s] | Voxel size [mm] | No. voxels | Time points |
|---|---|---|---|---|---|---|---|---|---|
| FCP | Resting state | Beijing | N/A | 198 | 3 | 2 | 3.1 × 3.1 × 3.6 | 64 × 64 × 33 | 225 |
| | Resting state | Cambridge, US | N/A | 198 | 3 | 3 | 3 × 3 × 3 | 72 × 72 × 47 | 119 |
| NKI | Resting state | Orangeburg, US | N/A | 30 | 3 | 1.4 | 2 × 2 × 2 | 112 × 112 × 64 | 404 |
| | Resting state | Orangeburg, US | N/A | 30 | 3 | 0.645 | 3 × 3 × 3 | 74 × 74 × 40 | 900 |
| CRIC | Resting state | Cambridge, UK | N/A | 73 | 3 | 2 | 3 × 3 × 3.8 | 64 × 64 × 32 | 300 |
| neuRosim | Resting state | (Simulated) | N/A | 100 | NA | 2 | 3.1 × 3.1 × 3.6 | 64 × 64 × 33 | 225 |
| NKI | Checkerboard | Orangeburg, US | 20 s off + 20 s on | 30 | 3 | 1.4 | 2 × 2 × 2 | 112 × 112 × 64 | 98 |
| | Checkerboard | Orangeburg, US | 20 s off + 20 s on | 30 | 3 | 0.645 | 3 × 3 × 3 | 74 × 74 × 40 | 240 |
| BMMR | Checkerboard | Magdeburg | 12 s off + 12 s on | 21 | 7 | 3 | 1 × 1 × 1 | 182 × 140 × 45 | 80 |
| CRIC | Checkerboard | Cambridge, UK | 16 s off + 16 s on | 70 | 3 | 2 | 3 × 3 × 3.8 | 64 × 64 × 32 | 160 |
| CamCAN | Sensorimotor | Cambridge, UK | Event-related | 200 | 3 | 1.97 | 3 × 3 × 4.44 | 64 × 64 × 32 | 261 |

For the enhanced NKI data, only scans from release 3 were used. Out of the 46 subjects in release 3, scans of 30 subjects were taken. For the rest, at least 1 scan was missing. For the BMMR data, there were 7 subjects at 3 sessions, resulting in 21 scans. For the CamCAN data, 200 subjects were considered only
*FCP* Functional Connectomes Project, *NKI* Nathan Kline Institute, *BMMR* Biomedical Magnetic Resonance, *CRIC* Cambridge Research into Impaired Consciousness, *CamCAN* Cambridge Centre for Ageing and Neuroscience

external toolbox SPMd generated them[35]. It provided a lot of information, which paradoxically could have limited its popularity. We believe that task fMRI analyses would strongly benefit if AFNI, FSL, and SPM provided some basic diagnostic plots. This way the investigator would be aware, for example, of residual autocorrelated noise in the GLM residuals. We provide a simple MATLAB tool (GitHub: plot_power_spectra_of_GLM_residuals. m) for the fMRI researchers to check if their analyses might be affected by imperfect pre-whitening.

To conclude, we showed that AFNI and SPM tested with option FAST had the best whitening performance, followed by FSL and SPM. Pre-whitening in FSL and SPM left substantial residual autocorrelated noise in the data, primarily at low frequencies. Though the problems were most severe for short TRs, different fMRI protocols were affected. We showed that the residual autocorrelated noise led to heavily confounded first level results. Low-frequency boxcar designs were affected the most. Due to better whitening performance, it was much easier to distinguish the assumed true experimental design from the assumed wrong experimental designs with AFNI and FAST than with FSL and SPM. This suggests superior specificity-sensitivity trade-off resulting from the use of AFNI's and FAST noise models. False negatives can occur when the design is event-related and there is residual positive autocorrelated noise at high frequencies. In our analyses, such false negatives propagated to the group level both when using a summary statistic model and a mixed effects model, although only to a small extent. Surprisingly, pre-whitening-induced false positives did not propagate to the group level when using the mixed effects model 3dMEMA. Our results suggest that 3dMEMA makes very little use of the standard error maps and does not differ much from the SPM's summary statistic model.

Results derived from FSL could be made more robust if a different autocorrelation model was applied. However, currently, there is no alternative pre-whitening approach in FSL. For SPM, our findings support more widespread use of the FAST method.

## Methods

**Data**. In order to explore a range of parameters that may affect autocorrelation, we investigated 11 fMRI datasets (Table 1). These included resting state and task studies, healthy subjects and a patient population, different TRs, magnetic field strengths, and voxel sizes. We also used anatomical MRI scans, as they were needed for the registration of brains to the MNI atlas space. Functional Connectomes Project (FCP)[36], NKI[37], and CamCAN data[38] are publicly shared anonymized data. Data collection at the respective sites was subject to their local institutional review boards (IRBs), who approved the experiments and the dissemination of the anonymized data. For the 1000 FCP, collection of the Beijing data was approved by the IRB of State Key Laboratory for Cognitive Neuroscience and Learning, Beijing Normal University; collection of the Cambridge data was approved by the

Massachusetts General Hospital partners' IRB. For the Enhanced NKI Rockland Sample, collection and dissemination of the data were approved by the NYU School of Medicine IRB. For the analysis of an event-related design dataset, we used the CamCAN dataset (Cambridge Centre for Ageing and Neuroscience, www.cam-can.org). Ethical approval for the study was obtained from the Cambridgeshire 2 (now East of England - Cambridge Central) Research Ethics Committee. The study from Magdeburg, "BMMR checkerboard"[39], was approved by the IRB of the Otto von Guericke University. The study of CRIC was approved by the Cambridge Local Research Ethics Committee (99/391). In all studies all subjects or their consultees gave informed written consent after the experimental procedures were explained. One rest dataset consisted of simulated data generated with the neuRosim package in R[40]. Simulation details can be found in Supplementary Information.

**Analysis pipeline**. For AFNI, FSL, and SPM analyses, the preprocessing, brain masks, brain registrations to the 2 mm isotropic MNI atlas space, and multiple comparison corrections were kept consistent (Fig. 4). This way we limited the influence of possible confounders on the results. In order to investigate whether our results are an artifact of the comparison approach used for assessment, we compared AFNI, FSL, and SPM by investigating (1) the power spectra of the GLM residuals, (2) the spatial distribution of significant clusters, (3) the average percentage of significant voxels within the brain mask, and (4) the positive rate: proportion of subjects with at least one significant cluster. The power spectrum represents the variance of a signal that is attributable to an oscillation of a given frequency. When calculating the power spectra of the GLM residuals, we considered voxels in native space within the same brain mask for AFNI, FSL, and SPM. For each voxel, we normalized the time series to have variance 1 and calculated the power spectra as the square of the discrete Fourier transform. Without variance normalization, different signal scaling across voxels and subjects would make it difficult to interpret power spectra averaged across voxels and subjects.

Apart from assuming dummy designs for resting state data as in recent studies[15,21,20], we also assumed wrong (dummy) designs for task data, and we used resting state scans simulated using the neuRosim package in R[40]. We treated such data as null data. For null data, the positive rate is the familywise error rate, which was investigated in a number of recent studies[15,21,20]. We use the term "significant voxel" to denote a voxel that is covered by one of the clusters returned by the multiple comparison correction.

All the processing scripts needed to fully replicate our study are at https://github.com/wiktorolszowy/fMRI_temporal_autocorrelation. We used AFNI 16.2.02, FSL 5.0.10, and SPM 12 (v7219).

**Preprocessing**. Slice timing correction was not performed as part of our main analysis pipeline, since for some datasets the slice timing information was not available. In each of the three packages we performed motion correction, which resulted in six confounders that we considered in the consecutive statistical analysis. As the 7T scans from the "BMMR checkerboard" dataset were prospectively motion corrected[41], we did not perform motion correction on them. The "BMMR checkerboard" scans were also prospectively distortion corrected[42]. For all the datasets, in each of the three packages we conducted high-pass filtering with frequency cutoff of 1/100 Hz. We performed registration to MNI space only within FSL. For AFNI and SPM, the results of the multiple comparison correction were registered to MNI space using transformations generated by FSL. First, anatomical scans were brain extracted with FSL's brain extraction tool[43]. Then, FSL's boundary-based registration was used for registration of the fMRI volumes to the anatomical scans. The anatomical scans were aligned to 2 mm isotropic MNI space using affine registration with 12 degrees of freedom. The two transformations were then combined for each subject and saved for later use in all analyses,

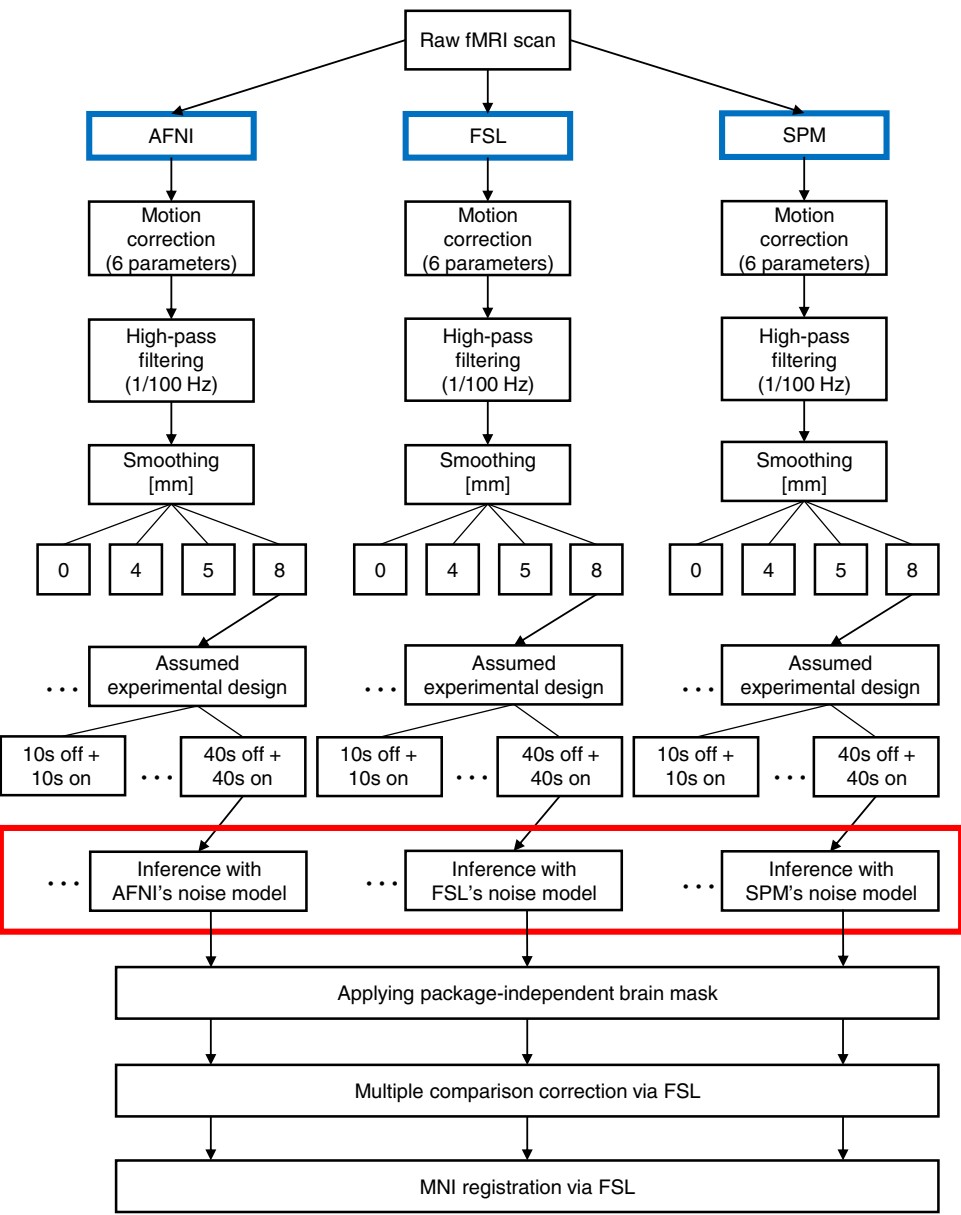

**Fig. 4** The employed analyses pipelines. For SPM, we investigated both the default noise model and the alternative noise model: FAST. The noise models used by AFNI, FSL, and SPM were the only relevant difference (marked in a red box)

including in those started in AFNI and SPM. Gaussian spatial smoothing was performed in each of the packages separately.

**Statistical analysis**. For analyses in each package, we used the canonical hemo-dynamic response function (HRF) model, also known as the double gamma model. It is implemented the same way in AFNI, FSL, and SPM: the response peak is set at 5 s after stimulus onset, while the post-stimulus undershoot is set at around 15 s after onset. This function was combined with each of the assumed designs using the convolution function. To account for possible response delays and different slice acquisition times, we used in the three packages the first derivative of the double gamma model, also known as the temporal derivative. We did not incorporate physiological recordings to the analysis pipeline, as these were not available for most of the datasets used.

We estimated the statistical maps in each package separately. AFNI, FSL, and SPM use restricted maximum likelihood (ReML), where autocorrelation is estimated given the residuals from an initial OLS model estimation. The ReML procedure then pre-whitens both the data and the design matrix, and estimates the model. We continued the analysis with the statistic maps corresponding to the $t$-test with null hypothesis being that the full regression model without the canonical HRF explains as much variance as the full regression model with

the canonical HRF. All three packages produced brain masks. The statistic maps in FSL and SPM were produced within the brain mask only, while in AFNI the statistic maps were produced for the entire volume. We masked the statistic maps from AFNI, FSL, and SPM using the intersected brain masks from FSL and SPM. We did not confine the analyses to a gray matter mask, because autocorrelation is at strongest in gray matter[4]. In other words, false positives caused by imperfect pre-whitening can be expected to occur mainly in gray matter. By default, AFNI and SPM produced $t$-statistic maps, while FSL produced both $t$- and $z$-statistic maps. In order to transform the $t$-statistic maps to $z$-statistic maps, we extracted the degrees of freedom from each analysis output.

Next, we performed multiple comparison correction in FSL for all the analyses, including for those started in AFNI and SPM. First, we estimated the smoothness of the brain-masked four-dimensional residual maps using the smoothest function in FSL. Knowing the DLH parameter, which describes image roughness, and the number of voxels within the brain mask (VOLUME), we then ran the cluster function in FSL on the $z$-statistic maps using a cluster defining threshold of 3.09 and significance level of 5%. This is the default multiple comparison correction in FSL and it refers to one-sided testing. Finally, we applied previously saved MNI transformations to the binary maps which were showing the location of the significant clusters.

## Code availability

All the processing scripts needed to fully replicate our study are at https://github.com/wiktorolszowy/fMRI_temporal_autocorrelation.

## Data availability

FCP[36], NKI[37], and CamCAN data[38] are publicly shared anonymized data. CRIC and BMMR scans can be obtained from us upon request. The simulated data can be generated again using our GitHub script simulate_4D.R.

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

## Acknowledgements

We would like to thank Michał Kosicki, Paul Browne, Anders Eklund, Thomas Nichols, Karl Friston, Richard Reynolds, Carsten Allefeld, Paola Finoia, Adrian Carpenter, Alison Sleigh, Gang Chen, and Guillaume Flandin for much valuable advice. Furthermore, we would like to thank the James S. McDonnell Foundation for funding the image acquisitions of the Cambridge Research into Impaired Consciousness (CRIC) group, and the CRIC group for sharing their data. Oliver Speck, Michael Hoffmann, and Aini Ismafairus Abd Hamid from the Otto von Guericke University provided us with the 7T data. We also thank the Neuroimaging Informatics Tools and Resources Clearinghouse and all of the researchers who have contributed with data to the 1000 Functional Connectomes Project and to the Enhanced Nathan Kline Institute - Rockland Sample. The informatics platform used for our data analyses was funded under an MRC research infrastructure award (MR/M009041/1). G.B.W. and J.A. acknowledge support from the EPSRC Centre for Mathematics in Healthcare (EP/N014588/1). W.O. was in receipt of scholarships from the Cambridge Trust and from the Mateusz B. Grabowski Fund. Also, W.O. was supported by the Guarantors of Brain.

## Author contributions

W.O., J.A. and G.B.W. designed the study; W.O. conducted the study; W.O., J.A., C.R. and G.B.W. analyzed the data; W.O. wrote the paper.

## Additional information

**Competing interests:** The authors declare no competing interests.

