## [Transparent Peer Review File · Nature Communications]

Reviewers' comments:

Reviewer #1 (Remarks to the Author):

Review: Accurate autocorrelation modeling substantially improves fMRI reliability

The present manuscript addresses important issues regarding the reliability of statistical parametric mapping in fMRI. I am convinced that this submission could be of high interest for the human brain mapping community and would help to advance the field. While this submission has the potential to contribute significantly to ongoing debates on fMRI methodology, there are a couple of issues that should be addressed before it would be suitable for publication (in particular for the interdisciplinary readership of this journal).

Strengths of this submission:

- Extensive dataset encompassing various fMRI experiments from different research sites and simulations
- Use of various established data processing and analysis methods/frameworks
- Code sharing (analysis and preprocessing)

Major Issues:

First of all, the authors address methodological issues that have already been raised in the seminal papers by Eklund et al. It is highly commendable that the authors use a more exhaustive (maybe even more realistic) dataset than Eklund et al. On the other hand, it would be very good to see if their analyses can replicate Eklund's findings and how different autocorrelation modeling approaches affect the overall results.

The authors should spend more time discussing and illustrating the differences of the employed autocorrelation models, in particular FAST, which is fairly unknown.

While false positives are an important concern, the authors do not properly address the risk of potential false negatives. This should be tested more rigorously.

Why did the authors use three different preprocessing and analysis pipelines when they were actually mainly interested in the performance of the different autocorrelation modeling approaches? This could introduce more variability and thus bias their results.

Preprocessing did not include slice timing correction, which is a quite established preprocessing step. Since it affects the time series (temporal smoothing due to interpolation) this step could have an influence on the overall results, which should be investigated in the data where slice timing information were available.

Minor Issues:

Please explicitly explain why the the variance was normalized when calculating the power spectra.

Sometimes (e.g., figure 2) it is not clear why particular parameters were selected and how they relate to other configurations that were not reported. While I appreciate the high complexity of the dataset and analyses, it would be really helpful if the authors could discuss how the reported findings generalize to the whole dataset.

Color figures (figure 3, S4) would be helpful.

Reviewer #2 (Remarks to the Author):

Summary:

This paper addresses a significant problem in fMRI data analysis methodology, namely, the impact of unaccounted-for autocorrelation in fMRI time series. This work is timely given the recent concerns about the accuracy and reliability of statistical analyses in fMRI, and the broader concerns about reproducibility and reliability in science. This paper analyzes the shortcomings of popular approaches, and takes the constructive step of identifying existing methods that can solve the problem. It also provides a diagnostic tool to assess the extent of the problem in previously analyzed data sets. The work seems very thorough, as the authors analyzed nearly 1000 data sets to arrive at their conclusions.

Although the approach is prosaic in some respects, in that no novel methodology is developed, I think this is a reflection of the nature of the problem. It is an unexpected result, because the problem of temporal autocorrelation in fMRI time series has probably been considered a "solved" problem. To the contrary, the authors show that, in practice, not all solutions are equal in performance; the more commonly-used solutions seem to perform the worst, while existing but less-commonly used approaches perform well. A major limitation of the work is that it focuses on first-level analyses. The effect on group-level inferences was not studied, which limits the potential significance of the result. I believe that this is something that the authors could readily address. The implication of this paper is that the high error rates reported in Eklund (PNAS, 2016) could come from residual temporal autocorrelation that is not accounted for in popular methods, but that is easily remedied using less-common existing methods. If that is the case, it would be a very important result. I truly hope the authors follow this story to the end by characterizing the effect on the group level.

Major Comments:

1. My main criticism is that the influence of poorly controlled temporal autocorrelation on group-level analyses was not characterized, limiting the potential significance and impact of this work. The supplementary materials briefly describe analyses comparing the false positive rates for a random effects analysis using either SPM or FAST, which seems to confirm the assertion made in the main text that temporal autocorrelation would not affect this type of analysis. But as the authors themselves say, the group-level mixed effects analysis is where temporal autocorrelation would likely make a significant difference. The mixed-effects analyses rely on the parameter variances estimated at the first level, which would be under-estimated if the temporal autocorrelations were not accounted for properly. Although it would entail more work, an analysis of 3dMEMA or FLAME under different first-level temporal autocorrelation processing schemes would be extremely valuable. Without such an analysis, we cannot know how much of an impact, if any, more accurate temporal autocorrelation analysis would have on the accuracy of group-level inferences that form the bulk of fMRI research. As I mention above, the implication of the present work is that accurate estimates of temporal autocorrelation might address the problems reported in Eklund (PNAS, 2016). If that were the case, it would be a huge result, and is something the authors should pursue.

Minor Comments:

1. On Page 1, line 32, the authors cite Purdon and Weisskoff (1998) in regards to the "AR(1) plus white noise" model used in SPM. Purdon and Weisskoff have nothing to do with what is implemented in SPM, and I imagine, might not want their paper to be incorrectly associated with that specific implementation. Perhaps Penny et al (2011) might be a better citation for this point. On the other hand, for the statement on Page 1, lines 7-10, "If this autocorrelation is not accounted for..." Purdon and Weisskoff (1998) might be a highly appropriate paper to cite.

2. On Page 4, lines 223-226, the author state "As the assumed design was a wrong design, a low power spectrum at the true design frequency suggests too strong pre-whitening, during which negative autocorrelations can be introduced. " Is this true? If the true design alternates at 1/24 Hz, but the assumed design is at a different frequency, the power attributable to the (unknown) true design will be in the residuals, which in turn will be accounted for by the temporal autocorrelation model and subsequent whitening. So, in this way, if the modeling/whitening is doing its job, the "true" design frequency will be suppressed. So my interpretation of this figure is that, in this scenario, the autocorrelation modeling and whitening are actually working correctly.

3. On Pages 6 through 8, lines 352 through 366, the authors attempt to provide an intuitive explanation of why lower assumed design frequencies result in increasing numbers of false positives. I understand the desire to provide an intuitive explanation that is accessible to non-statistical audiences. But I think the authors fall short here, and the explanation they provide seems too convoluted, and ends up being unnecessarily difficult to follow. A quite parsimonious explanation can be made using statistical principles. Auto-correlated processes have increasing variances at lower frequencies (or longer time scales). Thus, when the frequency of the design decreases, the mismatch between the true auto-correlated residual variance, and the incorrectly-estimated white noise variance grows. In this mismatch, the variance is under-estimated, resulting in a larger number of false positives.

Reviewers' comments:**Reviewer #1 (Remarks to the Author):****Review: Accurate autocorrelation modeling substantially improves fMRI reliability**

The present manuscript addresses important issues regarding the reliability of statistical parametric mapping in fMRI. I am convinced that this submission could be of high interest for the human brain mapping community and would help to advance the field. While this submission has the potential to contribute significantly to ongoing debates on fMRI methodology, there are a couple of issues that should be addressed before it would be suitable for publication (in particular for the interdisciplinary readership of this journal).

Strengths of this submission:

- Extensive dataset encompassing various fMRI experiments from different research sites and simulations
- Use of various established data processing and analysis methods/frameworks
- Code sharing (analysis and preprocessing)

Thanks a lot for the time you spent on reading and for the very interesting comments! The revised manuscript refers to your comments. Changes to the previous version are marked.

Major Issues:

First of all, the authors address methodological issues that have already been raised in the seminal papers by Eklund et al. It is highly commendable that the authors use a more exhaustive (maybe even more realistic) dataset than Eklund et al. On the other hand, it would be very good to see if their analyses can replicate Eklund's findings and how different autocorrelation modeling approaches affect the overall results.

There are three papers of Anders Eklund that were fundamental for our study:

1. Eklund, Anders, et al. "Does parametric fMRI analysis with SPM yield valid results?—An empirical study of 1484 rest datasets." *NeuroImage* 61.3 (2012): 565-578.
2. Eklund, Anders, et al. "Empirically investigating the statistical validity of SPM, FSL and AFNI for single subject fMRI analysis." *Biomedical Imaging (ISBI), 2015 IEEE 12th International Symposium on*. IEEE, 2015.
3. Eklund, Anders, Thomas E. Nichols, and Hans Knutsson. "Cluster failure: why fMRI inferences for spatial extent have inflated false-positive rates." *Proceedings of the National Academy of Sciences* (2016): 201602413.

The main finding of Eklund et al. 2012 was that the pre-whitening performance in SPM deteriorates with shorter TR. We wrote: *"The highly significant responses for the NKI datasets are in line with Eklund et al. (2012), where it was shown that for fMRI scans with short TR it is more likely to detect significant activation. The NKI scans that we considered had TR of 0.645s and 1.4s, in both cases much shorter than the usual repetition times"*. In addition to Eklund et al. 2012, we found the same relationship to be true for the AFNI's pre-whitening method and the current manuscript says: *"For short TRs, AFNI's performance deteriorated too, as autocorrelation spans much more than one TR and an ARMA(1,1) noise*

model can only partially capture it". For FSL, this relationship is not very strong, which could be explained with the high flexibility of the FSL's pre-whitening approach and two TRs (3s and 1.5s) that the method was calibrated to in Woolrich et al. 2001 "*Temporal autocorrelation in univariate linear modeling of FMRI data*".

One of the main findings of Eklund et al. 2015 was that familywise error rates (FWERs) in FSL are higher than in SPM, while FWERs for SPM are slightly higher than for AFNI. We also observed lowest FWERs for AFNI: "*Interestingly, in Eklund et al. 2015 AFNI, FSL and SPM were already compared in the context of first level fMRI analyses. AFNI resulted in substantially lower false positive rates than FSL and slightly lower false positive rates than SPM. We also observed lowest false positive rates for AFNI. Opposed to Eklund et al. 2015, which compared the packages in their entirety, we compared the packages only with regard to pre-whitening.*"

Eklund et al. 2016 pointed to problems in some analyses which make multiple testing correction with the cluster inference. Most worrying problems were reported when using the cluster defining threshold (CDT) of 0.01. In our study we opted to use only one multiple testing correction procedure: FSL's default cluster inference with CDT 0.001 and significance level of 5%. We expect that some results in Eklund et al. 2016 would have been less problematic if there had been less bias in the standard error maps. Two datasets used in our study were also used in Eklund et al. 2016 ("*FCP Beijing*" and "*FCP Cambridge*"). Like for other datasets, we observed strong residual autocorrelated noise in the GLM residuals using both FSL's and SPM's pre-whitening methods also for these two datasets.

Eklund et al. 2015 and Eklund et al. 2016 compared the packages in their entirety. While their approach pinpointed a number of problems, it makes the evaluation of single processing steps in the analysis more difficult and, as a result, a direct comparison to our results is not straightforward. Perhaps the most relevant conclusion of Eklund's two previous studies might be that many statistical methods used in task fMRI analyses are not working as well as it had been assumed for years.

Finally, and in response to the second reviewer's comments on group studies, we have now performed 1-sample t-test analyses with SPM's random effects model and with AFNI's mixed effects model on all the datasets. To make a comparison with Eklund et al. 2016 more straightforward, we considered sample sizes of 20, just like in analyses summarized in Fig. 1 of Eklund et al. 2016. Our resulting work led to the results section "*Group studies*" (pages 7-9), Fig. 4, Supplementary Tables 2-4 and discussion on page 10. More information about relationship of our work to Eklund et al. 2016 can be found in our response to the first comment of the second reviewer (pages 12-13 of this document).

As the above comparison of our work to Anders Eklund's studies could confuse some readers of the main part of the paper, we would be glad if, in the case of a publication, the reviewers wanted to publish our correspondence alongside the paper, as is the case with some papers published in this journal.

The authors should spend more time discussing and illustrating the differences of the employed autocorrelation models, in particular FAST, which is fairly unknown.

We tried to keep the manuscript concise. However, another problem is the lack of literature describing the different methods. For example, there is not any paper or other documentation describing the SPM's default pre-whitening exactly in the way it is used today. Regarding FAST, the first paper describing the approach appeared after we submitted the paper. In the revised manuscript we refer to this work ("*Accurate modeling of temporal correlations in rapidly sampled fMRI time series*", Nadege Corbin et al. 2018). AFNI's pre-whitening approach is not described in any paper, though the method is very straightforward. For FSL, there is a dedicated paper of Woolrich et al. (2001): "*Temporal autocorrelation in univariate linear modeling of FMRI data*", but the method was updated after the publication of the paper. For example, the parameter of the Tukey taper used to smooth the spectral density estimates was originally fixed to 15, while in current FSL logs one can observe different values of this parameter. The value depends now on the length of the time series.

We have extended the introduction to include more information on the methods.

While false positives are an important concern, the authors do not properly address the risk of potential false negatives. This should be tested more rigorously.

This is a very good point. In our study we evaluated primarily false positives as there is a more reliable ground truth. For example for a flickering checkerboard experimental design, significant activations found in CSF/WM or in GM, but for a completely different assumed experimental design than the true one, could be treated as false positives. Yet, the full extent of the active site in V1 for this experiment (or any other task fMRI experiment) is unknown. For four datasets with flickering boxcar checkerboard tasks, we observed relatively similar significant activation in visual cortex across AFNI/FSL/SPM/FAST pre-whitening algorithms, cf. Fig. 3 (in manuscript) and Supplementary Fig. 1.

However, sensitivity losses might be expected when the design is event-related and when at high frequencies there is residual positive autocorrelated noise (power spectra above 1). This causes the activation and the rest periods not to be well decorrelated. This is shown and discussed in section "*Event-related design studies*" on page 7 (together with Supplementary Fig. 4). With increasing autocorrelation, the coefficients in the regression model go to zero (in the limit). We have acknowledged the previous limitations of the paper in the discussion by adding "*False negatives can occur when the design is event related and there is residual positive autocorrelated noise at high frequencies. In our analyses, such false negatives propagated to the group level both when using a random effects model and a mixed effects model, although only to a small extent.*".

Why did the authors use three different preprocessing and analysis pipelines when they were actually mainly interested in the performance of the different autocorrelation modeling approaches? This could introduce more variability and thus bias their results.

We minimized variability in the software packages to negligible levels, so that pre-whitening is the only relevant difference. Preprocessing was not exactly the same, as high-pass filtering/detrending is applied both to the data and to the model, and the use of the very same high-pass filter would require hard-coding in two packages. Then, the possible resulting numerical problems could have confounded the results. However, motion correction was performed in each of the packages using the same number of parameters (six), high-pass

filtering employed the same frequency cut-off (1/100 Hz), while the spatial smoothing applied the same sizes of the kernel. All in all, we expect the confounding effect of the slightly different preprocessing to be negligible. Importantly, the brain mask, the multiple comparison correction, and the MNI registration were kept exactly the same across the AFNI/FSL/SPM pipelines, cf. Fig. 1 and:

https://github.com/wiktorolszowy/fMRI_temporal_autocorrelation/blob/master/multiple_comparison_correction_and_registration_to_MNI.R

Preprocessing did not include slice timing correction, which is a quite established preprocessing step. Since it affects the time series (temporal smoothing due to interpolation) this step could have an influence on the overall results, which should be investigated in the data where slice timing information were available.

Following your suggestion, we reanalyzed the “*CRIC checkerboard*” and “*CamCAN sensorimotor*” datasets. For the “*CRIC checkerboard*” dataset, slice timing information was included in the scans’ headers, while for the “*CamCAN sensorimotor*” dataset, slice timing information was given in the paper describing the dataset (Shafto et al. 2014 “*The Cambridge Centre for Ageing and Neuroscience (Cam-CAN) study protocol: a cross-sectional, lifespan, multidisciplinary examination of healthy cognitive ageing.*”). Scripts which we used for this additional analysis are at:

https://github.com/wiktorolszowy/fMRI_temporal_autocorrelation/tree/master/slice_timing_correction

The revised manuscript contains the new results section “*Slice timing correction*” (page 7). Page 10 discusses the results.

Minor Issues:

Please explicitly explain why the the variance was normalized when calculating the power spectra.

Power spectrum at a given frequency represents the variance of the time series attributable to an oscillation of this frequency. If the overall variance of the time series is not normalized, it will be difficult to average power spectra across voxels or subjects. The following MATLAB example confirms that power spectra indeed depend on the overall variance of the time series:

```
>> mean((abs(fft(randn(200, 1)))).^2)/200)
ans =
    0.9822

>> mean((abs(2*fft(randn(200, 1)))).^2)/200)
ans =
    3.7700
```

Initially, we did not put this comment to the manuscript as it is a very technical issue and variance normalization was used without explanation for example in Eklund et al. 2015

“Empirically investigating the statistical validity of SPM, FSL and AFNI for single subject fMRI analysis”. Following your comment, we added the following text to the methods section: *“Without variance normalization, different signal scaling across voxels and subjects would make it difficult to interpret power spectra averaged across voxels and subjects”*.

Sometimes (e.g., figure 2) it is not clear why particular parameters were selected and how they relate to other configurations that were not reported. While I appreciate the high complexity of the dataset and analyses, it would be really helpful if the authors could discuss how the reported findings generalize to the whole dataset.

We did consider different levels of spatial smoothing, different assumed experimental designs and different types of data (simulated rest, acquired rest and acquired task). This way we tested if the results were an artefact of the employed parameters. Importantly, differences between AFNI/FSL/SPM/SPM+FAST were consistent across different smoothing levels, different assumed experimental designs and different types of data. The highlighted parameters were chosen to represent a broad range of typical use cases, and this is made clear in the text. Further results are available from the GitHub repository and these are now referenced in the text (page 7):

https://github.com/wiktorolszowy/fMRI_temporal_autocorrelation/tree/master/figures

On the next pages you can see the power spectra of the GLM residuals for smoothing of 4 mm and for 8 mm, as well as for the assumed boxcar design of 10s off + 10s on and for the assumed boxcar design of 40s off + 40s on. In all these four scenarios, pre-whitening from FSL and SPM led to substantial residual positive autocorrelated noise at low frequencies, while FAST pre-whitening led to most flat power spectra. Figs. 5-6 show power spectra for the task datasets tested with true designs. Again, FSL and SPM pre-whitening methods led to much positive autocorrelated noise left at low frequencies. Manuscript's Fig. 3 and supplement show that these consistent differences between AFNI/FSL/SPM/SPM+FAST exist with regard to first level results too (Supplementary Figs. 1, 2, 3). All in all, we consider the results from the main part of the manuscript to be generalizable.

Figure 1 Power spectra of the GLM residuals for assumed experimental boxcar design 10s off + 10s on and for spatial smoothing of 4 mm

Figure 2 Power spectra of the GLM residuals for assumed experimental boxcar design 10s off + 10s on and for spatial smoothing of 8 mm. This figure is in the manuscript (Fig. 2).

Figure 3 Power spectra of the GLM residuals for assumed experimental boxcar design 40s off + 40s on and for spatial smoothing of 4 mm.

Figure 4 Power spectra of the GLM residuals for assumed experimental boxcar design 40s off + 40s on and for spatial smoothing of 8 mm.

Figure 5 Power spectra of the GLM residuals for true experimental designs and for spatial smoothing of 4 mm.

Figure 6 Power spectra of the GLM residuals for true experimental designs and for spatial smoothing of 8 mm.

Color figures (figure 3, S4) would be helpful.

We chose black-white-gray palette in Fig. 3 and Supplementary Figure 4 as we did not want to suggest these are group level results. Such black-white-gray figures depicting spatial distribution of significant clusters overlaid on each other from different runs were used in the supplement of Eklund et al. 2016 (Fig. S18, page 25):

<http://www.pnas.org/content/pnas/suppl/2016/06/27/1602413113.DCSupplemental/pnas.1602413113.sapp.pdf>

Reviewer #2 (Remarks to the Author):

Summary:

This paper addresses a significant problem in fMRI data analysis methodology, namely, the impact of unaccounted-for autocorrelation in fMRI time series. This work is timely given the recent concerns about the accuracy and reliability of statistical analyses in fMRI, and the broader concerns about reproducibility and reliability in science. This paper analyzes the shortcomings of popular approaches, and takes the constructive step of identifying existing methods that can solve the problem. It also provides a diagnostic tool to assess the extent of the problem in previously analyzed data sets. The work seems very thorough, as the authors analyzed nearly 1000 data sets to arrive at their conclusions.

Although the approach is prosaic in some respects, in that no novel methodology is developed, I think this is a reflection of the nature of the problem. It is an unexpected result, because the problem of temporal autocorrelation in fMRI time series has probably been considered a “solved” problem. To the contrary, the authors show that, in practice, not all solutions are equal in performance; the more commonly-used solutions seem to perform the worst, while existing but less-commonly used approaches perform well. A major limitation of the work is that it focuses on first-level analyses. The effect on group-level inferences was not studied, which limits the potential significance of the result. I believe that this is something that the authors could readily address. The implication of this paper is that the high error rates reported in Eklund (PNAS, 2016) could come from residual temporal autocorrelation that is not accounted for in popular methods, but that is easily remedied using less-common existing methods. If that is the case, it would be a very important result. I truly hope the authors follow this story to the end by characterizing the effect on the group level.

Thanks a lot for the time you spent on reading and for the very interesting comments! The revised manuscript refers to your comments. Changes to the previous version are marked.

Major Comments:

1. My main criticism is that the influence of poorly controlled temporal autocorrelation on group-level analyses was not characterized, limiting the potential significance and impact of this work. The supplementary materials briefly describe analyses comparing the false positive rates for a random effects analysis using either SPM or FAST, which seems to confirm the assertion made in the main text that temporal autocorrelation would not affect this type of analysis. But as the authors themselves say, the group-level mixed effects analysis is where temporal autocorrelation would likely make a significant difference. The mixed-effects analyses rely on the parameter variances estimated at the first level, which would be under-estimated if the temporal autocorrelations were not accounted for properly. Although it would entail more work, an analysis of 3dMEMA or FLAME under different first-level temporal autocorrelation processing schemes would be extremely valuable. Without such an analysis, we cannot know how much of an impact, if any, more accurate temporal autocorrelation analysis would have on the accuracy of group-level inferences that form the bulk of fMRI research. As I mention above, the implication of the present work is that accurate estimates of temporal autocorrelation might address the problems reported in Eklund (PNAS, 2016). If that were the case, it would be a huge result, and is something the authors should pursue.

Bias in mixed effects analyses resulting from non-white noise at the first level was reported in Bianciardi et al. 2004 “*Evaluation of mixed effects in event-related fMRI studies: impact of first-level design and filtering*”. If the group analysis employs standard error maps and these are biased, one would expect biased/inaccurate group analysis results. Following your suggestion, we investigated this issue further with AFNI’s 3dMEMA. To make our group analyses more comparable to results presented in Eklund et al. 2016 (addressing the first comment of the first reviewer), for each dataset and each assumed experimental design, we ran 1-sample t-tests on 20 subjects, both with SPM’s summary statistic approach and with 3dMEMA. The corresponding scripts are on GitHub:

https://github.com/wiktorolszowy/fMRI_temporal_autocorrelation/blob/master/make_group_analyses_random_effects.m

https://github.com/wiktorolszowy/fMRI_temporal_autocorrelation/blob/master/make_group_analyses_mixed_effects.R

https://github.com/wiktorolszowy/fMRI_temporal_autocorrelation/blob/master/make_combined_results_from_group_runs.R

Our resulting work led to the results section “*Group studies*” (pages 7-9), Fig. 4, Supplementary Tables 2-4 and discussion on page 10. Overall, we did not show that better pre-whitening would improve results from Eklund et al. 2016 a lot, at least not for the SPM’s summary statistic approach and not for 3dMEMA. However, the most worrying results from Eklund et al. 2016 were for CDT of 0.01, which we did not consider. For multiple comparison correction, we only used CDT of 0.001, as this is now the most standard approach (e.g. default in FSL and SPM).

In our analyses, 3dMEMA results were surprisingly similar to results for the SPM’s summary statistic approach. As we observed for 3dMEMA a strange negative relationship between the magnitude of the t-statistic map and the amount of significant activation, we think the method

does not always work as well as it was shown in the simulations in Chen et al. 2012 “*fMRI group analysis combining effect estimates and their variances*”. In fact, Chen et al. 2012 compared 3dMEMA with FLAME and found lower FWER for 3dMEMA than for FLAME. This conflicts with Eklund et al. 2016 (cf. Fig. 1, Eklund et al. 2016). We discussed our results with Gang Chen, the author of 3dMEMA, and currently we try to understand with Gang Chen why the use of 3dMEMA led to such surprising results (both in our work resulting from your comment, and in Eklund et al. 2016). However, we think further work on group analyses could be beyond the scope of this paper. Mixed effects models can be considered more optimal than random effects models as they make use of more information. Even if better pre-whitening does not lead to better specificity with one of the currently available mixed effects models, bias in standard error maps should be avoided.

In our study we showed that imperfect pre-whitening can lead to higher sensitivity for event-related designs. This leads to higher regression coefficients and affects group analyses both when using random and mixed effects models. However, main results in Eklund et al. 2016 refer to false positives, so in our study it was only 3dMEMA where better pre-whitening could have led to statistically better results than those that were presented in Eklund et al. 2016. We think we were not able to find higher specificity for better pre-whitening approaches, as it seems to us now that 3dMEMA is not an ideal group analysis technique.

Importantly, in the recent follow-up study of Eklund et al.: “*Cluster failure revisited: Impact of first level design and data quality on cluster false positive rates*” (2018) it was shown that cleaning the data using ICA FIX improves specificity a lot. Such an approach corresponds to more accurate pre-whitening, although this was not mentioned in Eklund et al. 2018. The new version of the manuscript refers to Eklund et al. 2018 (page 10).

Minor Comments:

1. On Page 1, line 32, the authors cite Purdon and Weisskoff (1998) in regards to the “AR(1) plus white noise” model used in SPM. Purdon and Weisskoff have nothing to do with what is implemented in SPM, and I imagine, might not want their paper to be incorrectly associated with that specific implementation. Perhaps Penny et al (2011) might be a better citation for this point. On the other hand, for the statement on Page 1, lines 7-10, “If this autocorrelation is not accounted for...,” Purdon and Weisskoff (1998) might be a highly appropriate paper to cite.

Indeed, Purdon and Weisskoff 1998 did not suggest exactly the same pre-whitening as the one which is used in SPM now. For example SPM estimates the noise model only based on voxels which seem to be active when no pre-whitening is conducted ($z_{stat} > 3.1$) and the specific implementation of the Restricted Maximum Likelihood has changed several times, among others between v6906 and v7219. However, Friston et al. 2000 “*To Smooth or Not to Smooth? Bias and Efficiency in fMRI Time-Series Analysis*” says: “*Purdon and Weisskoff (1998) suggested the use of an AR(1) plus white noise model.*”. Furthermore, Friston et al. 2002 “*Classical and Bayesian Inference in Neuroimaging: Applications*” says “*These bases were chosen given the popularity of AR plus white noise models in fMRI (Purdon and Weisskoff, 1998).*” As Friston et al. 2002 describes the current SPM’s pre-whitening approach quite accurately, we are now citing this work instead of Purdon and Weisskoff 1998. Also, following your suggestion we added Purdon and Weisskoff 1998 to the intuitive explanation why pre-whitening matters.

2. On Page 4, lines 223-226, the author state “As the assumed design was a wrong design, a low power spectrum at the true design frequency suggests too strong pre-whitening, during which negative autocorrelations can be introduced. “ Is this true? If the true design alternates at 1/24 Hz, but the assumed design is at a different frequency, the power attributable to the (unknown) true design will be in the residuals, which in turn will be accounted for by the temporal autocorrelation model and subsequent whitening. So, in this way, if the modeling/whitening is doing its job, the “true” design frequency will be suppressed. So my interpretation of this figure is that, in this scenario, the autocorrelation modeling and whitening are actually working correctly.

Indeed, if the pre-whitening procedure was perfect, all signal at frequencies unrelated to the experiment would be removed. However, the pre-whitening algorithms were not developed to deal with such strong “noise”, so that assuming wrong design to task data and looking at the power spectra could be treated as an implicit way of investigating sensitivity. Nevertheless, it is only an implicit way.

The differences in the amount of significant activation for the true design between AFNI/FSL and SPM/FAST for this dataset (“*BMMR checkerboard*”, TR=3s) were surprisingly large. Lenoski et al. 2008 “*On the performance of autocorrelation estimation algorithms for fMRI analysis*” described how inflexible methods (like global noise models) could lead to negative autocorrelations being introduced during pre-whitening for long TRs. Given the differences between AFNI/FSL and SPM in power spectra for the wrong designs, we initially thought that SPM’s pre-whitening was the culprit. However, in the meantime we investigated this dataset further and found that a lot of significant activation was found for SPM in the motion regressors, which were considered as confounders in the GLM analysis. As our statistical inference was based on t-test on the canonical function (rather than F-test on all regressors), SPM led to very little significant activation in our original analysis. Motion correction algorithms from AFNI and FSL did not lead to significant activation in the motion regressors. It is surprising given the study Oakes et al. 2005 “*Comparison of fMRI motion correction software tools*”, where it was shown that different motion correction algorithms lead to only negligibly different analysis results. We contacted the authors of the “*BMMR checkerboard*” study (Aini Ismafairus Abd Hamid and Oliver Speck), who made us aware of a recent paper describing the problem which we encountered: Yakupov et al. 2017 “*False fMRI Activation After Motion Correction*”. The authors found that SPM’s motion correction works much less accurately than AFNI and FSL for the special case of ultra high field data and limited acquisition field of view, a situation which was not covered in Oakes et al. 2005. The “*BMMR checkerboard*” scans are ultra high field data and were made with a limited acquisition field of view.

Importantly, the “*BMMR checkerboard*” scans were prospectively motion corrected using the approach from Thesen et al. 2000 “*Prospective acquisition correction for head motion with image-based tracking for real-time fMRI*”. We originally conducted motion correction on the “*BMMR checkerboard*” scans, since no motion correction is perfect and we wanted to keep the processing pipeline the same across all the eleven datasets. Given the problems with SPM’s motion correction for the special case of ultra high field data and limited acquisition field of view, and the fact that prospective motion correction was performed on this dataset, we reran all AFNI/FSL/SPM/FAST analyses for the “*BMMR checkerboard*” dataset without additional motion correction. Now there are much more similar results across the different pre-whitening algorithms for the “*BMMR checkerboard*” dataset. The new version of the manuscript shows the updated analysis.

The “*BMMR checkerboard*” dataset was the only dataset where scans were prospectively motion corrected. As performing double motion correction is not a straightforward analysis step, we do not refer in the current version of the manuscript to our previous problems with motion correction in SPM. Such an explanation could confuse the reader as the study is about pre-whitening rather than preprocessing. However, in the case of a publication, we would be glad if the reviewers wanted to publish our correspondence alongside the paper, as is the case with some papers published in this journal.

3. On Pages 6 through 8, lines 352 through 366, the authors attempt to provide an intuitive explanation of why lower assumed design frequencies result in increasing numbers of false positives. I understand the desire to provide an intuitive explanation that is accessible to non-statistical audiences. But I think the authors fall short here, and the explanation they provide seems too convoluted, and ends up being unnecessarily difficult to follow. A quite parsimonious explanation can be made using statistical principles. Auto-correlated processes have increasing variances at lower frequencies (or longer time scales). Thus, when the frequency of the design decreases, the mismatch between the true auto-correlated residual variance, and the incorrectly-estimated white noise variance grows. In this mismatch, the variance is under-estimated, resulting in a larger number of false positives.

We did change the description accordingly.

REVIEWERS' COMMENTS:

Reviewer #1 (Remarks to the Author):

I highly appreciate the authors' efforts to improve this manuscript. While I feel that my concerns have been addressed thoroughly from a technical perspective, I have concerns that the relevant information might not be clear enough for the target audience. This could lead to confusion in the neuroimaging community. Again, I have no concerns about the technical aspects of this submission and I am impressed by the authors' extensive analyses and truly believe that their findings are relevant for the community. However, this would be the right time to address the presentation of their results and improve the didactic value of the manuscript.

I am certain that this submission to this prestigious interdisciplinary journal will stimulate discussions and the authors can make sure to drive the debate in the right direction from the start. Maybe I have been unclear about this, so I will try to be more precise. As a reader I would like to know:

Can Eklund's observations (particularly PNAS 2016) be (in part) attributed to improper autocorrelation modeling?

What is the theoretical explanation for this?

How can this problem be solved or at least reduced?

Are there any undesired consequences of an improved AR modeling approach?

I fully appreciate that the authors already address most of these issues in their initial submission. However, I would be very grateful if they could discuss these topics more concretely and to the point. However, if the editor and the authors believe that there is no more room for improvement on this level, I am holding my peace.

Reviewer #2:

[Was not available to re-review. Reviewer #3 agreed to comment on the authors' response to Reviewer #2's points from the previous round]

Reviewer #3 (Remarks to the Author):

The authors did a nice job of investigating the impact of different strategies in dealing with the temporal correlation embedded in the residuals of the fMRI model at the individual level. The investigation was quite thorough and impressive. I only have a couple of minor suggestions as below.

I appreciate that the authors examined the effect of modeling the temporal correlation structure at the individual level on the group level. However, the exploration was not as thorough as the individual level, which is completely understandable in the current context. First of all, the term of "random-effects" analysis at the group level may be popular in one particular package, but such a usage does not accurately capture the nature of the model. The typical group-level model is GLM (or one-sample t-test in the case of the manuscript), which simply ignores the measurement errors of effect estimates from the individual subjects. There is no random-effects term in the model; therefore, naming it as a random-effect models is misleading. The authors should simply call it GLM or t-test.

Secondly, the mixed-effects modeling approach weighs the effect estimates from the individual subjects. Those weights are the reciprocals of the sums of two variance components: cross-subjects variance and within-subject variances. While the first component is constant (but a priori unknown), the second component varies across subjects. The GLM approach simply assumes that the within-subject variances are the same for all subjects or are very small relative to the first component (cross-subjects variance). The report of mixed-effects modeling results "surprisingly

similar to" those from GLM is actually not surprising at all: it simply means that 1) there was not much variability among those within-subject variances, or 2) those within-subject variances are relatively small compared to the cross-subjects variance. It seems that the few datasets the authors investigated for the task-related scenarios do not actually paint an exhaustive picture about the impact of within-subject variances on the group results unless the authors are willing to go the extra mile exploring the detailed profiles of those within-subject variances relative to the cross-subjects variance. Even then you might still not be able to cover a large portion of the scenarios. My personal opinion is that numerical simulations would actually offer a more complete and accurate perspective than a few real datasets could reveal. In other words, the few datasets adopted in the current manuscript do not necessarily provide a fair demonstration for the potential advantages of mixed-effects modeling. It would be more appropriate for the authors to at least mention this limitation of their current study.

In addition, the report of "strange negative relationship between the magnitude of t-statistic maps and the amount of significant activation" from the mixed-effects modeling is not strange either. Again, the weights involved in the mixed-effects modeling are the combinations of those two variance components, resulting in an intricate nonlinear relationship. When artificially changing the magnitude of the t-statistics, you not only changed the relative magnitudes between the two variance components, but also might have magnified the relative impact of any outlying subjects in the group. In that case, it may require some special outlier modeling strategy. I believe such a complexity was partially explored in Chen et al. (2012). It is beyond the scope of the current manuscript to exhaustively investigate the various scenarios and to provide insightful results, so I would simply remove this whole "strange relationship" part that was resulted from the artificial manipulation of t-statistics.

Lastly, regarding "AFNI's pre-whitening approach is not described in any paper": The algorithm was published in the Appendix of Chen et al. (2012), *NeuroImage* 60(1):747-765, which might be better cited in the Introduction?

REVIEWERS' COMMENTS:**Reviewer #1 (Remarks to the Author):**

Thanks a lot for the very interesting comments!

I highly appreciate the authors' efforts to improve this manuscript. While I feel that my concerns have been addressed thoroughly from a technical perspective, I have concerns that the relevant information might not be clear enough for the target audience. This could lead to confusion in the neuroimaging community. Again, I have no concerns about the technical aspects of this submission and I am impressed by the authors' extensive analyses and truly believe that their findings are relevant for the community. However, this would be the right time to address the presentation of their results and improve the didactic value of the manuscript.

I am certain that this submission to this prestigious interdisciplinary journal will stimulate discussions and the authors can make sure to drive the debate in the right direction from the start. Maybe I have been unclear about this, so I will try to be more precise. As a reader I would like to know:

Can Eklund's observations (particularly PNAS 2016) be (in part) attributed to improper autocorrelation modeling?

What is the theoretical explanation for this?

How can this problem be solved or at least reduced?

Eklund's observations from 2016 can be attributed to improper autocorrelation modeling only to a very small extent. Eklund et al. 2016 referred to group level analyses, while in our analyses we found that poor pre-whitening primarily leads to problems at the single subject level. We observed that group level analyses had slightly lower sensitivity following poor pre-whitening, but Eklund et al. 2016 analysed specificity only, so we could not refer to that study. Eklund et al. 2016 identified problems related to the modeling of the spatial autocorrelation function. In the current study, we identified problems related to the temporal autocorrelation modeling. Our manuscript refers to the other studies (Eklund et al. 2012 and Eklund et al. 2015), which are strongly related to the current study (also single subject analyses). In particular, our study is an extension of the analyses from Eklund et al. 2012 and Eklund et al. 2015.

Now the discussion section includes: *“Interestingly, a recent report suggested that the FSL's tool ICA FIX applied to task data can successfully remove most of the physiological noise (Eklund et al. 2018). This was shown to lower the familywise error rate at the group level compared to previous findings (Eklund et al. 2016). Such an approach corresponds to more accurate pre-whitening. However, in our analyses the different pre-whitening methods affected the group level analyses only in a very negligible way. While Eklund et al. 2016 found that group level analyses are strongly confounded by spatial autocorrelation modeling, we found that single subject analyses were strongly confounded by the pre-whitening accuracy.”*

Are there any undesired consequences of an improved AR modeling approach?

Perhaps slightly lower sensitivity for some experiments with boxcar designs, but only at the cost of deteriorated specificity. Sensitivity can be improved with accurate HRF modeling, for example using the temporal derivative. However, the temporal derivative increases sensitivity only if the statistical inference is based on a test which includes the temporal derivative (F-test needed rather than t-test). This is rarely the case, as the temporal derivative is used in most studies only as a confounder, for example when using the default FSL processing pipeline. This is beyond the scope of this paper.

I fully appreciate that the authors already address most of these issues in their initial submission. However, I would be very grateful if they could discuss these topics more concretely and to the point. However, if the editor and the authors believe that there is no more room for improvement on this level, I am holding my peace.

Reviewer #2:

[Was not available to re-review. Reviewer #3 agreed to comment on the authors' response to Reviewer #2's points from the previous round]

Reviewer #3 (Remarks to the Author):

Thanks a lot for the very interesting comments!

The authors did a nice job of investigating the impact of different strategies in dealing with the temporal correlation embedded in the residuals of the fMRI model at the individual level. The investigation was quite thorough and impressive. I only have a couple of minor suggestions as below.

I appreciate that the authors examined the effect of modeling the temporal correlation structure at the individual level on the group level. However, the exploration was not as thorough as the individual level, which is completely understandable in the current context. First of all, the term of “random-effects” analysis at the group level may be popular in one particular package, but such a usage does not accurately capture the nature of the model. The typical group-level model is GLM (or one-sample t-test in the case of the manuscript), which simply ignores the measurement errors of effect estimates from the individual subjects. There is no random-effects term in the model; therefore, naming it as a random-effect models is misleading. The authors should simply call it GLM or t-test.

The name “*random effects model*” is used in SPM, which is why we previously used this term. Now we replaced it with “*summary statistic model*”.

Secondly, the mixed-effects modeling approach weighs the effect estimates from the individual subjects. Those weights are the reciprocals of the sums of two variance components: cross-subjects variance and within-subject variances. While the first component is constant (but a priori unknown), the second component varies across subjects. The GLM approach simply assumes that the within-subject variances are the same for all subjects or are very small relative to the first component (cross-subjects variance). The report of mixed-effects modeling results “surprisingly similar to” those from GLM is actually not surprising at all: it simply means that 1) there was not much variability among those within-subject variances, or 2) those within-subject variances are relatively small compared to the cross-subjects variance. It seems that the few

datasets the authors investigated for the task-related scenarios do not actually paint an exhaustive picture about the impact of within-subject variances on the group results unless the authors are willing to go the extra mile exploring the detailed profiles of those within-subject variances relative to the cross-subjects variance. Even then you might still not be able to cover a large portion of the scenarios. My personal opinion is that numerical simulations would actually offer a more complete and accurate perspective than a few real datasets could reveal. In other words, the few datasets adopted in the current manuscript do not necessarily provide a fair demonstration for the potential advantages of mixed-effects modeling. It would be more appropriate for the authors to at least mention this limitation of their current study.

We have commented on this in the discussion section now. While simulations are a great way of investigating the performance of a statistical test (ground truth is known), our findings on real data are interesting too. We see them as complementary to the simulation findings presented in Chen et al. 2012.

In addition, the report of “strange negative relationship between the magnitude of t-statistic maps and the amount of significant activation” from the mixed-effects modeling is not strange either. Again, the weights involved in the mixed-effects modeling are the combinations of those two variance components, resulting in an intricate nonlinear relationship. When artificially changing the magnitude of the t-statistics, you not only changed the relative magnitudes between the two variance components, but also might have magnified the relative impact of any outlying subjects in the group. In that case, it may require some special outlier modeling strategy. I believe such a complexity was partially explored in Chen et al. (2012). It is beyond the scope of the current manuscript to exhaustively investigate the various scenarios and to provide insightful results, so I would simply remove this whole “strange relationship” part that was resulted from the artificial manipulation of t-statistics.

We do not find this analysis crucial for our study, so we removed it now.

Lastly, regarding “AFNI’s pre-whitening approach is not described in any paper”: The algorithm was published in the Appendix of Chen et al. (2012), NeuroImage 60(1):747-765, which might be better cited in the Introduction?

We did not know such a description was in the Appendix of Chen et al. 2012. Now there is a proper citation in the introduction.